# Exploring Perceptions of a Fresh Food Prescription Program during COVID-19

**DOI:** 10.3390/ijerph191710725

**Published:** 2022-08-28

**Authors:** Rachel Zimmer, Ashley Strahley, Jane Weiss, Sheena McNeill, Allison S. McBride, Scott Best, David Harrison, Kimberly Montez

**Affiliations:** 1Department of Gerontology and Geriatric Medicine, Wake Forest University School of Medicine, Winston Salem, NC 27157, USA; 2Department of Public Health Sciences, Wake Forest University School of Medicine, Winston Salem, NC 27157, USA; 3BestHealth, Atrium Health Wake Forest Baptist, Winston Salem, NC 27157, USA; 4Department of Pediatrics, Wake Forest University School of Medicine, Winston Salem, NC 27157, USA; 5H.O.P.E. of Winston Salem, Winston Salem, NC 27106, USA; 6New Communion, Winston Salem, NC 27105, USA

**Keywords:** food insecurity, qualitative research, food prescription, produce prescription, food access, older adults, social determinants of health, COM-B model, wellness, health promotion

## Abstract

This qualitative study aimed to elicit the perspectives of individuals with food insecurity (FI) who were enrolled in a Fresh Food Prescription (FFRx) delivery program through a collaboration between an academic medical center and multiple community partners in the southeastern United States. Semi-structured interviews and open-ended survey responses explored the experiences of participants enrolled in a FFRx delivery program during the COVID-19 pandemic. The interviews probed the shopping habits, food security, experience, and impact of the program on nutrition, health, and well-being; the surveys explored the perceptions of and satisfaction with the program. A coding scheme was developed inductively, and a thematic analysis was conducted on raw narrative data using Atlas.ti 8.4 to sort and manage the data. The themes included that the program promoted healthy dietary habits, improved access to high-quality foods, improved well-being, enhanced financial well-being, and alleviated logistical barriers to accessing food and cooking. Participants provided suggestions for FFRx improvement. Future studies may facilitate improved clinical–community partnerships to address FI.

## 1. Introduction

During the height of the COVID-19 pandemic, the Census Household Pulse Survey (2020) revealed that 23% of surveyed households in the United States (U.S.) experienced food insecurity (FI), or the unreliable access to sufficient affordable, quality, and nutritious food; Black and Latinx individuals reported disproportionately higher rates of FI, at 36% and 32%, respectively, versus 18% for white individuals. [1,2] In 2020, the prevalence of FI in North Carolina, a southeastern state of the U.S. was significantly higher than the national average of 10.5% (12.1%) [1]. Food insecurity is a growing health problem in the US, with an estimated $52 billion spent on healthcare due to FI [3,4]. Feeding America has identified racism as a main driver of FI in the U.S. [1,3,5].

Food insecurity in children is associated with a two-times higher reporting of fair or poor health [6], and in older adults with higher rates of reported fair or poor health as well as adverse health outcomes [6,7,8,9]. Individuals with FI make tradeoff decisions about whether to buy food or other basic needs, and their dietary intake tends to be lower in quality and more calorically dense [10]. Specifically, disability, lack of home ownership, poverty, low median income, and unemployment were identified as the five additional main drivers of FI [3].

The COVID-19 pandemic has contributed to the public health crisis of FI in the U.S. due to the increased demand and economic hardship wrought by the pandemic, with food banks worldwide increasing their reach by 132% [11]. The pandemic has also inspired a variety of produce prescription programs. However, evidence surrounding the impacts of produce prescription programs is still being researched, with 85% of produce prescription studies having been published over the last 4 years (12). Participant input is integral to help guide future interventions that address social needs, and there is a need for qualitative studies that capture participant feedback on how they were impacted by social programs, such as food prescription programs, during the pandemic [12,13]. 

The aims of this study are to (1) understand the impact of a food prescription program on food access, eating habits, and health behaviors before and during the program’s implementation, and (2) to explore participants’ acceptability and satisfaction with this pilot produce prescription program that was developed within a U.S. academic–community partnership. 

## 2. Theoretical Justification

The Behavior Change Wheel (BCW) is a framework for designing and evaluating programs [14]. For this pilot produce prescription program, we used the BCW to guide both the produce prescription intervention development and interview questions (14). At the center of the BCW core is Capabilities (C), Opportunities (O), Motivation (M), and Behaviors (COM-B), which helped identify components that influence behavior and accounted for factors outside an individual that make a behavior possible [14] (Figure 1). The premise of the COM-B model is that behaviors are influenced primarily by capability, opportunity, and motivation (green ring). The red ring is comprised of nine different intervention functions that can be used to influence one or more of the underlying functions of behavior [14]. Lastly, the gray outer ring are policy categories that can help to support the interventions used to support behavior change. 

For example, according to the COM-B model, to change behavior, such as increasing healthy eating habits, one or more of the COM-B components would need to be changed or improved upon and addressed. We used COM-B to guide the implementation of the Fresh Food Prescription Program (FFRx) and help to influence and facilitate healthy eating behaviors by supporting positive intervention targets, such as reducing barriers around the transportation and cost of obtaining healthy foods in low-income households by environment restructuring through the delivery of fresh produce each week [14] (Table 1). The COM-B model has been used in several qualitative studies to evaluate the impacts of dietary interventions, and COM-B has proven to be a useful tool to inform these interventions on how to promote healthy behavior [15,16]. The Theoretical Domains Framework (TDF) helps to further elaborate on the COM-B model by describing the behavioral determinants that can be mapped onto the COM-B components, further explaining domains that are specific to target behaviors [17]. Several qualitative studies use the COM-B model and TDF to explore the facilitators and barriers to dietary behavior change.

## 3. Materials and Methods

### 3.1. Design

This qualitative study was nested inside a larger mixed methods study that explored the impacts of the FFRx program on the health of our participants. The quantitative data from this mixed methods study were previously published [18,19]. Semi-structured interviews were used to elicit dialogue from program participants to evaluate the impact of a community-informed FFRx program on dietary behaviors. [18,19]. The questions were based on guidance using the COM-B model TDF (Table 2).

Additionally, subsets of FFRx participants were surveyed using open-ended questions as a part of the FFRx intervention for quality improvement purposes by FFRx staff bi-monthly. The answers to the survey questions enabled the operations team and community partners to gain real-time input from participants to incrementally improve the program’s aspects. For example, if participants consistently reported wanting a specific produce item, the farmers included this produce item in the upcoming deliveries (Table 3).

### 3.2. Study Setting and Population

We conducted this study at a major academic medical center in Winston-Salem within Forsyth County, the third-largest urban area in North Carolina (NC), which is a southeastern state of the U.S. In Winston-Salem, 16% of all households have FI, impacting 21% of households with children (19). Participants were recruited from a convenience sample of individuals who received home-delivered produce and meals for at least three months from the FFRx program. The inclusion criteria for study participation were: (1) 18 years of age or older; (2) a current or past participant in the FFRx program; and (3) English or Spanish as the preferred language. 

### 3.3. Program Description

The FFRx program was a delivery program implemented to address COVID-19′s impact on food access for individuals living in under-resourced communities [20]. The program was developed after gaining community input through focus groups and included weekly delivery of regionally-sourced produce and pre-packaged meals to individuals who self-identified as having FI [18,19]. Individuals with FI were referred to the program by social workers working with the medical center’s house call program, doctor’s offices, local faith communities, and non-profit partners [21]. Participants who declined to participate in the data collection were not denied participation in the FFRx program.

This FFRx program was a partnership between an academic medical center, CBOs, local farmers, the local food bank, and faith communities within the U.S. Community partners who helped implement FFRx were identified through existing relationships and by their shared mission for addressing FI in the community. The regional produce and pre-packaged meals were provided weekly at no cost to the participants from March 2020 to October 2021. Produce boxes included newsletters and recipes that were developed weekly by a nutritionist. Additionally, the nutritionist provided individual coaching via phone for participants interested in reaching their health goals. 

### 3.4. Data Collection

#### 3.4.1. Interviews

The interview guide was based on the COM-B framework and was developed from a detailed review of the literature as well as previous focus groups that were performed by the FFRx team prior to the FFRx intervention; the guide explored participants’ experiences with food, including access, eating habits, health behaviors, acceptability, and affordability [18] (Table 2). The interview guide was reviewed and modified through an iterative process. Research staff members trained in qualitative interview techniques conducted the interviews via telephone. Spanish-language interviews were conducted by a certified bilingual staff member. Semi-structured interviews were conducted via telephone in English (n = 15) and Spanish (n = 3) during the months of October–December 2020. Consent was obtained via the telephone. All interviews were digitally recorded, transcribed verbatim in English or Spanish, and verified against the audio recording for accuracy. Spanish transcriptions were translated into English by a third-party certified translation service. The interviews continued until thematic saturation was reached, or no new information, categories, or themes emerged [22].

#### 3.4.2. Surveys

FFRx program staff also conducted bi-monthly surveys with program beneficiaries for quality improvement (Table 3). The survey respondents were asked to share their favorite and least favorite produce and meal items from each week’s delivery, their suggestions on additional produce items to include in the boxes, input on produce items they had difficulty preparing, and the participants’ perception of what the most impactful part of the FFRx program has been for them. A convenience sample of participants were selected for each survey, with an average of 12 participants per survey (range 7–18 participants per survey). The surveys were conducted by telephone with only English-speaking program participants between December 2020 and June 2021. Eleven total bi-monthly surveys were conducted, garnering a total of 137 responses. The data were transcribed verbatim during the call. Participants were eligible to be sampled multiple times. 

### 3.5. Analysis

Digital transcripts from the interviews were reviewed and organized using a text-based analysis software program, Atlas.ti version 8.4 (Scientific Software Development GmbH, Berlin, Germany). A thematic analysis was conducted on the interview data, following the principles outlined by Green and Thorogood [23,24]. Data analysts first reviewed the interview transcripts and inductively developed a codebook by identifying relevant codes based on the interview data. Two researchers independently coded the transcripts and then discussed these with the research team after each coding session to resolve any coding discrepancies until full agreement was reached. The study team reviewed sections of the coded transcript to uncover themes and chose direct quotations that highlighted these themes. 

Data analysts also developed a codebook for coding survey responses related to the impact of the FFRx program through open coding of the responses. The survey data were coded in Microsoft Excel by a single analyst. Once coding was complete, the data were synthesized by code. The remaining open-ended survey data were synthesized by question, without formal coding. The study team reviewed sections of the coded transcript to uncover themes and chose direct quotations that highlighted these themes. 

This study was approved by the Wake Forest University School of Medicine’s Institutional Review Board (IRB58931).

## 4. Results

### 4.1. Participant Characteristics

Of the 150 FFRx participants, 18 participated in in-depth interviews between October–December 2020. Most of the interview participants were female (83%) and 60 years of age or older (61%), self-identified as Black or non-Hispanic (72%), and did not have children in the home (56%). Additionally, a total of 97 participants were surveyed during the FFRx intervention. Of the 97 survey participants, 78.4% (n = 76) self-identified as Black and 99% (n = 135) were age 60 or older. The characteristics of survey respondents in Table 4 also note the demographics of the total number of survey responses, as some participants completed multiple surveys over time (Table 4).

### 4.2. Key Themes Identified

The interview conversations were centered around participant experiences with food access, nutrition, health, and impacts of the FFRx program. We identified four major themes that were relevant to participant experiences with the FFRx program: (1) participants were motivated to eat or prepare healthy foods; (2) participants felt that the FFRx positively impacted aspects of their well-being; (3) participants felt that the FFRx alleviated logistical barriers to food access; and (4) participants provided suggestions for program improvement. Among these main themes, we identified several sub-themes that are supported by representative quotations (Table 5).

#### 4.2.1. Participants Were Motivated to Eat or Prepare Healthy Foods

Participants listed a variety of reasons for why they chose to eat healthily or prepare healthy foods at home. About half of interview participants said that they were motivated to eat healthily to maintain or improve their health, including losing weight or maintaining a healthy weight, or managing a health condition, such as diabetes or high cholesterol. “I try to watch my cholesterol. I do have high cholesterol. I try to maintain more fruits and vegetables into my diet and grilled foods.” (Non-Hispanic White (NHW) Female, Age 64).

Other motivators included enjoying the taste of fresh foods, past exposure to a variety of fresh foods, as well as information from the media that promoted healthy eating. One participant explained, “… sometimes it’s very tasty, the fresh food. When you prepare your freshly made salad, it’s delicious. With avocado, a little white rice… very tasty. Very good.” (Hispanic Female, Age 37).

While most participants were motivated to eat healthily, a few participants reported challenges to doing so, such as having children in the household who are picky eaters. Two participants said that they try to prepare vegetables in different ways to get their children to try them, and one said that she must do “a whole marketing [routine]” to get her children to try new things. (Black Female, Age 35).

#### 4.2.2. FFRx Positively Impacted Aspects of Participants’ Well-Being

The interview and survey participants were asked to describe how they or their family were impacted by receiving the produce boxes, including how the boxes changed their dietary habits. Most participants reported a positive impact on their health behaviors. We identified three subthemes within this primary theme.

#### 4.2.3. FFRx Promoted Healthy Dietary Habits 

Nearly all interview participants and a few survey participants said that they were eating fresh fruits and vegetables more frequently after receiving the FFRx produce boxes. Some respondents also said that the program encouraged them to be more conscientious of their food choices. One participant reported, “… before we are eating fruit and vegetables three times a week… Now we got this package with the vegetable—fresh fruit and vegetables, we are eating every day.” (AA Female, Age 32) Many participants reported that they were encouraged by the positive impacts the program had on their overall physical health. “I feel healthier with all this produce… My sugar levels have lowered. It used to be 300, now it’s under 200. (Native Hawaiian/Pacific Islander (NHW) Female, Age 64) Several participants reported that the program helped to increase their energy, while others appreciated the improvement in skin conditions, weight management, and lowered blood pressure.

A few interview participants also said that the produce boxes exposed the children in their household to new fruits and vegetables, and one participant said that her children were eating fresh fruit as a snack instead of candy. “It helped us a lot, because my children started consuming more… They stopped eating candy, things like that, taking fruit instead.” (Hispanic Female, Age 28).

Some interview participants reported that the weekly nutrition letter and recipes provided with each produce box were informative and helpful. One participant, who identified as Hispanic, and another, who identified as Ethiopian, said that the recipes helped them cook “American” food for their children, which the children enjoyed. Participants enjoyed the convenience of the recipes and felt that they helped to facilitate cooking.

#### 4.2.4. FFRx Improved Mental Health and Feelings of Isolation

Some participants noted that the social aspect of the FFRx, including regular contact with the delivery team, made them feel like someone cared about them. “Just knowing someone out there cares… It nice to have someone come to your door… Helps me feel not so alone.” (NHW Female, Age 64) Another participant noted that the program alleviated social isolation during the COVID-19 pandemic, “It gave me something to look forward to… to talk to some people.” (Black Male, Age 57).

#### 4.2.5. FFRx Enhanced Financial Well-Being 

Some interview and survey participants described overall FI prior to their participation in the program, saying that they worried about where their next meal would come from, would skip meals because they could not afford food, or would have to choose between medications and food (or forgo other necessities like cleaning supplies). Other participants said that their income and/or supplemental foods benefits did not allow for them to purchase “healthy” foods, such as fresh fruits and vegetables. One participant explained, “It offsets when I don’t have food or money to buy food. It [fills] that gap.” (Black Female, Age 76) While another participant noted that the FFRx program “… saves me money. It ensures that I will have at least 1 meal a day.” (Male, Unknown Race, Age 55).

#### 4.2.6. FFRx Alleviated Logistical Barriers to Accessing Food and Cooking 

Many interview and survey participants reported that the FFRx program alleviated both physical and logistical barriers, such as transportation, to accessing healthy food. Most of the FFRx participants noted that they appreciated the delivery of produce boxes versus having to pick up a box each week. Some participants specifically remarked on the importance of the delivery during the pandemic, when specific health issues and/or general concerns about COVID exposure made them cautious about going to public places, like the grocery store. One participant noted that the delivery aspects of FFRx, “Made my life better. Especially during the pandemic, with my autoimmune disorder, I couldn’t go out.” (Black Female, Age 70).

Participants reported that cooking skills were facilitated by recipes and tips provided in the weekly newsletter. However, some older adults reported that they no longer enjoyed cooking or were unable to cook due to physical limitations and appreciated the convenience of the prepared meals provided each week with the produce boxes. 

#### 4.2.7. Participants Provided Feedback and Suggestions for FFRx Program Improvements

Both interview and survey participants provided feedback and suggestions on the FFRx program. Most survey participants reported being able to eat more than half of the produce in their produce box each week, and very few reported any difficultly preparing the produce. Participants who had trouble preparing produce included not being able to cook because of arthritis or other physical limitations, or items (e.g., collard greens, carrots) being difficult to cut.

Several interview and survey participants commented on the freshness of the produce with one participant noting that the variety of produce helped her “come up with creative dishes”, and another noting that the produce was “fresher than at the store.” (Black Female, Age 35; Hispanic female, Age 37).

As described earlier, several interview participants reported that they found the nutrition letter and recipes provided as informative and helpful. A couple of participants suggested that they would like to see more health information included. Some participants suggested that the program include more culturally-diverse recipes (e.g., Hispanic, African, Israeli), recipes in Spanish, and familiar recipes with a twist (e.g., macaroni and cheese with a vegetable). 

## 5. Discussion

The findings of this qualitative study demonstrate that participants perceived the FFRx program, which was a partnership between an academic health system, CBOs, farmers, and faith communities within the U.S., as positively impacting their health and well-being while filling a critical need for access to healthy foods. Participants also provided suggestions for improvement, including more culturally effective strategies. 

This is the first theory-based qualitative study known to the authors that explored the impacts a FFRx delivery program had on participants with FI during the COVID-19 pandemic within the U.S. The results confirmed earlier reported barriers to healthy eating in low-income communities, including budget, transportation, convenience, and time [1,15,24]. Several authors note that programs such as Meals on Wheels, community feeding programs, and food pantries, fill a critical need as an emergency food source for people with FI and may also improve health. However, FI is a stigmatizing condition, and many people within the U.S. do not feel comfortable accessing existing resources for support [23,24,25,26,27]. 

Food assistance programs, specifically home-delivered interventions, can improve health outcomes, food security, and health care utilization [26,27,28,29,30]. Two studies by Berkowitz et al., found that medically tailored, home-delivered meals provided in the U.S. improved FI by some measures of dietary intake, health, and health utilization [27,28,31]. 

Within the U.S., there has been increasing interest to address FI in a meaningful way by health care providers by partnering with community-based organizations (CBOs), as well as a growing need for evidence to help guide the most impactful ways to do this [31,32,33]. Rates of patient engagement with community resources for FI are low after screening/referrals from clinical settings [25], highlighting the importance of clinical engagement in community-based assets to mitigate FI. This qualitative study demonstrates a culturally appropriate healthy food plan with the community to meet the needs of our patients who suffer with FI. 

Most of the participants served by the FFRx program were older adults and were unable to leave their homes during the pandemic due to the increased risk of illness and death. Since the same drivers delivered food each week, the regular contact from FFRx staff led to friendships among the FFRx team and participants, helping to alleviate social isolation and improving mental health. Participants and staff communicated that the social aspects this FFRx incorporated via delivery and check-in calls were very meaningful. Since FI increases the risk of depression in older adults by three-fold, the social aspects of food delivery programs have been stated by researchers to be as important as the provision of food itself [32,34,35,36]. Further research is needed to determine the impact of food prescription programs on mental health.

### 5.1. Strengths and Limitations

This is one of the first studies to use the COM-B model to both inform FFRx program development and qualitative interviews that complemented a previously published quantitative data study [18,19]. The nature of our qualitative interviews allows researchers to explore the results and impacts that a food prescription program has on participants that a quantitative design cannot fully explore [14]. 

This study had several limitations. First, participants were recruited from a single academic institution in the southeastern U.S., so results may not be transferable to other institutions. As well, participants who agreed to participate in an interview may not be representative of all FFRx participant perspectives. Another limitation is the overrepresentation of women in the sample, which may have influenced results, although women may also be the ones preparing meals for their households. 

Program participants who were English-speaking were eligible to be sampled in multiple surveys, and individual program participants were not identified in the survey dataset. Survey data on the impact of the FFRx program were presented according to the total number of survey responses and did not correspond to the number of participants. Therefore, a limitation of the survey data is that caution should be exercised in interpreting and reporting themes to avoid overstating the prevalence and salience of themes to program participants.

### 5.2. Implications for Health Policy

Produce and food prescription programs are a promising way for health care providers and CBOs to address social drivers of health in the U.S. In North Carolina, there have been several CBOs and health care providers who have been partnering on such programs for several years [37]. Additionally, the Harvard Center for Health Law and Policy, Feeding America, and the Rockefeller Foundation have all published reports evaluating and describing known and potential impacts of U.S. programs that address FI [37,38,39]. 

There are a variety of strategies used to address FI based on funding, partnerships, and resources, which provides a challenge when evaluating and sustaining such programs. [40] Payors in the US have not historically provided funding to support food access, but this has shifted over the last few years [37]. In North Carolina, FI is also being addressed in the context of Health Opportunities Pilots, a Medicaid initiative to “test evidence-based, non-medical interventions designed to reduce costs and improve health [41].” As future food prescription programs take shape, it is essential for meaningful partnerships between health care providers, CBOs, farmers, and participants to inform the infrastructure of these programs, accounting for and building on existing assets while considering the partner, community, and participant needs, and sustainability.

## 6. Conclusions

This qualitative study elicited participant perspectives to improve food prescription programming. We found that a food prescription delivery program during the COVID-19 pandemic helped participants feel healthier and overcome financial and logistical barriers to healthy eating. Future studies may facilitate improved clinical–community food prescription program interventions to mitigate FI among under-resourced populations. 

## Figures and Tables

**Figure 1 ijerph-19-10725-f001:**
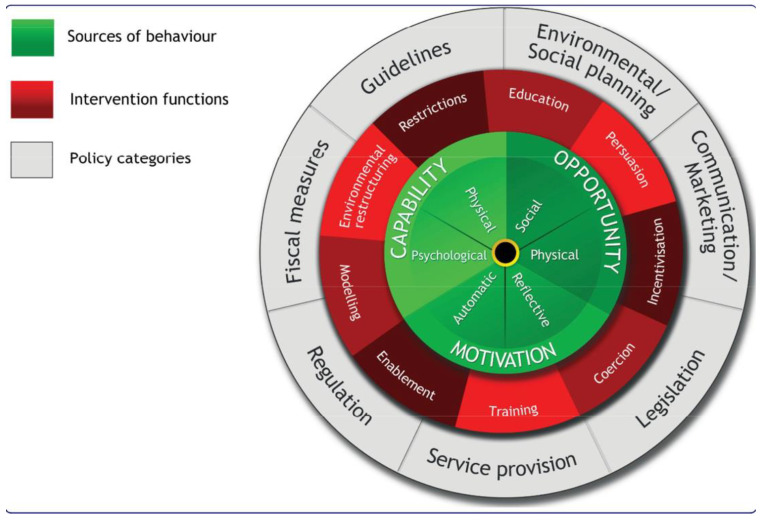
The Behavior Change Wheel (used with permission from Michie et al. [14]).

**Table 1 ijerph-19-10725-t001:** Mapping of the COM-B to TDF Domains and Strategies to Evaluate the Impacts of Interventions on Dietary Behaviors [17].

COM-B Domains		TDF Constructs	Goals/Barriers/Strategy
Capability	Psychological Capability	Knowledge Memory, attention, and decision processes	Goal: Improve knowledge about how to choose and prepare healthy foods. Barriers: Knowledge deficits/education *Strategies:* *Education:* Provide biweekly coaching with nutritionist through phone outreach; weekly tips through newsletters based on produce provided
	Physical Capability	Skills	Goal: Improve skills and capability to prepare produce provided in the program. Barriers: Functional or cognitive limitations. Lack of basic cooking skills. *Strategies:* *Education:* Individual counselling by dietician; Nutritionist guided recipes in newsletter *Enabling interventions:* Easy to prepare recipes. Screening for and provision of items needed to cook meals (microwaves, air fryers, utensils). Provision of durable medical equipment to support mobility for functionally frail. *Training:* Newsletter topics on label reading, cooking tricks
Opportunity	Physical Opportunity	Environmental context and resources	Goal: Improve access to healthy foods, increase daily consumption of produce. Barriers: Pandemic and financial strain effects on food access. Lack of in person opportunities for healthy habits education and support due to pandemic. *Strategies:* *Environmental change:* Provide access to locally grown U.S. produce through weekly produce delivery. Nutritionist guided prepackaged meals in partnership with Second Harvest Food Bank and Providence Kitchen provide heart healthy meals. *Education:* Phone consults with YMCA Health coach and nutritionist if requested by participant for nutritional guidance.
	Social Opportunity	Social influences	Goal: Establish a routine of integrating healthy foods into family and peer settings. Barriers: Social isolation. Differing cultural identities that impact types of preferred foods. *Strategies:* *Persuasion:* Use data from interviews of participants to tailor program to tastes by incorporating produce that people suggested that they prefer, including apples, berries, peaches, and potatoes. Eggs provided with each box per recommendation of participants. Peer support through health coaching.
Motivation	Reflective Motivation	Role and Identity Beliefs about Capabilities and Intentions	Goal: Increase motivation and self-efficacy when selecting, preparing, and consumption of healthy foods. Barriers: Negative self-talk, decreased perceived capabilities. Perceived lack of control over food access and over financial situation *Strategies:* *Enablement:* Provide deliveries of food boxes at same time each week with same driver for consistency. *Education:* Provision of education on healthy behaviors, benefits. *Incentivization:* Weekly contact with drivers and occasional treats in boxes based on participant input.
	Automatic Motivation	Optimism Reinforcement	Goal: Develop self-driven goals related to improving one’s own physical and mental health. Barriers: Lack of well-defined goals for health or motivation for eating healthier. Depression or mood disorders related to social isolation that can prevent motivation to care for oneself. *Strategies:* *Modelling:* Care coordinator called and remind participants of deliveries as well as provision of tips for using boxes each week. *Enablement:* Provide easy recipes to go along with produce.

**Table 2 ijerph-19-10725-t002:** Mapping of the COM-B to TDF Domains and Strategies to Develop the Semi-Structured Interview Guide.

Capabilities-Opportunities-Motivation (COM-B) Framework Domains	TDF Construct	Interview Questions
Psychological Capability	Knowledge, Memory, Attention, and Decision Processes	How (if at all) did the fresh food program change your shopping habits in relation to fresh foods?
		Before receiving the food prescription box, how many times per month would you buy fresh fruits and vegetables? Why?
		How (if at all) did receiving the fresh food box influence how often you eat fresh fruits and vegetables?
Physical Capability	Skills	How many times have you cooked the recipe in the newsletter?
		What other ways could this program help you cook the produce from the fresh food box?
Physical Opportunity	Environmental Context and Resources	Did you have everything you needed to cook the recipe provided? Prompt: ingredients, cooking equipment
		How easy or difficult is it for you to get fresh fruits and vegetables? What makes it hard for you to buy fresh fruits and vegetables? What would make it easier for you to buy fresh fruits and vegetables?
		Where do you usually go to get fresh fruits and vegetables?
Social Opportunity	Social Influences	Tell me about you or your family’s experience with the fresh food box you received each week.
		How (if at all) did your family benefit from the fresh food program?
Reflective Motivation	Role and Identity Beliefs about Capabilities and Intentions	What are the main reasons you like to shop where you do for fresh fruits and vegetables?
		What motivates you to eat healthy?
Program Quality and Feedback (not COM-B)		What did you like about the nutrition letter included in the box? What didn’t you like, or what content would you like to see in the nutrition letter ongoing?
		What were your opinions about the delivery of the fresh foods to your home? What went well with the delivery? What didn’t go well?

**Table 3 ijerph-19-10725-t003:** Survey Questions Asked of Participants of the Fresh Food Rx Program.

Survey Questions
Were you able to eat over ½ of the produce provided in the box this week?
What was your favorite produce item?
What was your least favorite produce item?
Which pre-packaged meal did you like the most?
What additional meals/produce items would you like to see included?
Did you have any difficulties preparing the produce or warming up the meals?
What is most impactful about receiving the Fresh Food Rx delivery each week?

**Table 4 ijerph-19-10725-t004:** Characteristics of Interview and Survey Participants.

Participant Characteristics	Interview participants N = 18 (%)	Survey Participants N = 97 (%)	Survey Participant Responses N = 137 (%)
*Race/Ethnicity*			
Non-Hispanic, Black	13 (72.2%)	76 (78.4%)	107 (78.1%)
Hispanic White	3 (16.7%)		
Non-Hispanic White	2 (11.1%)	17 (17.5%)	22 (16.1%)
Other	0 (0%)	4 (4.1%)	8 (5.9%)
*Sex*			
Female	15 (83.3%)	77 (79.4%)	107 (78.1%)
Male	3 (16.7%)	20 (20.6%)	30 (21.9%)
*Age*			
<60 (28–57)	7 (38.9%)	1 (1%)	2 (1.5%)
≥60 (60–88)	11 (61.1%)	96 (99%)	135 (98.5%)
*Household Composition*			
Children in Home	8 (44.4%)	2 (2%)	5 (3.6%)
No Children in Home	10 (55.6%)	9 (9.3%)	113 (82.5%)
Unknown		15 (15.5%)	19 (13.9%)

**Table 5 ijerph-19-10725-t005:** Key Themes and Illustrative Quotes of FFRx Interview Participants.

Theme	Subtheme	Selected Illustrative Quotes
Participants Were Motivated to Eat or Prepare Healthy Foods		*“… before we are eating fruit and vegetables three times a week, something like that, because it’s not enough. Now we got this package with the vegetable—fresh fruit and vegetables, we are eating every day.”*
		*“You know how we humans are. We got our own little routine and our own little way of doin’ things, but when you’re all sent me those boxes it made me to remember. It was on my mind. It was before my face. I went ahead and ate it.”*
		*“[My children] get the experience of seein’ different fruits, different greens that they may not have tried, normally. Like zucchini. It’s not that often that I go out and buy a zucchini… ”*
FFRx Positively Impacted Aspects of Wellbeing	FFRx Promoted Healthy Dietary Habits	*“I feel healthier with all this produce… My sugar levels have lowered. It used to be 300, now it’s under 200. I used to be stressed about can I eat a meal.”*
		*“The doctor is proud of my A1C and keeping my weight down.”*
		*“My doctor is pleased with my health because it’s stabilized.”*
	FFRx Positively Impacted Social Isolation	*“It gave me something to look forward to, meeting your delivery guys… it gave me a chance to talk to some people, to get out my room. They would tell us not to go out into the halls and circulate so much because of that COVID.”*
	FFRx Improved Financial Wellbeing	*“It helps me. You know I don’t get much in food stamps, so it helps me get through the rest of the month… It evens out.”*
		*“[I’m] able to purchase cleaning supplies and medications without giving up food. I used to skip meals because I could not afford it. That’s not good for being diabetic.”*
		*“It saves me money. It ensures that I will have at least 1 meal a day.”*
FFRx Alleviated Logistic Barriers to Food Access		*“I know you have made my life better. Especially during the pandemic, with my autoimmune disorder, I couldn’t go out… ”*
		*“For someone to be able to deliver something, helps me a lot… Someone older cannot get out and about like they want to… ”*
		*“Because I don’t do much cooking now… Sometimes I can get around, sometimes I can’t. By me having arthritis, it bothers me to get around to the store and the stove… ”*
Feedback and Suggestions for Improvement		*“ “I like the little pamphlets that’s in there… I like the information that it provides. The knowledge on different things, different ways you can cook somethin’.”*
		*“I like my own way of doin’ things. I like to cook my own style is what I’m sayin’. The ingredients and everything that were there, that were fine, but I like to cook the way I like to cook ’cause it’s the way I was raised… ”*

## Data Availability

The datasets generated during and/or analysed during the current study are available from the corresponding author on reasonable request.

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
