# Peer review of "Exploring Perceptions of a Fresh Food Prescription Program during COVID-19"

_ijerph, 2022, doi:10.3390/ijerph191710725_

Round 1

Reviewer 1 Report

The study exploring the impacts of a US produce prescription program on the participants, with the aims to (1) understand the impact of a food prescription program on food access, eating habits, and health behaviors before and during the program's implementation, (2) to explore participants' acceptability and satisfaction with this pilot produce prescription program that was developed within an US academic-community partnership. The methodological tools applied were semi-structured interviews and surveys. Data and information collected were processed through a coding scheme, and thematic analysis was conducted on raw narrative data using Atlas.ti 8.4 to sort and manage data.

Despite the intrinsic limitations described by the Authors, all understandable and shareable, the paper demonstrates that it has the methodological basis to achieve the objectives and represent an information source in the area under study. However, some descriptive deficiencies heavily undermine the scientific requirements.

Although "Abstract", "Introduction" and "Materials and Methods" anticipate that data and information will be subjected to analysis, the only data described concern the raw ones on interview participants, survey participants, their socio-demographic characteristics; no other data is reported in "Results".

Authors identified four major themes, and some corresponding subthemes, relevant to participant experiences with the produce prescription program, but do not indicate the frequencies of responses included in the themes. Likewise, there is no trace of the processing carried out with the Atlas.ti 8.4 software.

Unfortunately, these shortcomings undermine any scientific value of the paper. I highly recommend Authors to integrate the text by describing the missing numerical values.

To be corrected

Row 77: … TDF …

The acronym "TDF" is shown for the first time, without indicating the words of origin. It is necessary to report the words from which it originates.

Row 121: … the lead and senior author …

The identification criteria of those who developed the interview guide should be completed with specific skills, so that they can have value in materials and methods.

Rows 139-140: Surveys were conducted via telephone in English

The concept is repeated a few lines below

Row 174 and following: Table 4. Characteristics of Participants in Interviews and Surveys

Why was the statistical analysis not conducted to see if the samples of the interviewees in the two moments of the research are or not significantly different? I believe it is necessary to give a minimum of scientific consistency to the work.

Row 185 and following: Table 5. Key Themes and Illustrative Quotes.

How come the number of answers belonging to the key themes are not reported, nor any kind of statistical processing? This is substantial information for scientific purposes.

Reviewer 2 Report

The authors explore participants' views to improve a food prescription delivery program. They find that the program helped participants feel healthier and overcome financial and logistical barriers to healthy eating during the COVID-19 pandemic. The assessment is based on the Behavior Change Wheel (BCW).

The article is interesting. The issue is relevant and timely, and should be of interest to the journal’s readers. The paper is straightforward and well-written. Methods are appropriate. Figures and tables are informative and clearly presented. The results are interesting and are useful to improve the program. The discussion is supported by the results and the limitations of the qualitative study are acknowledged. However, I would suggest better integrating the BCW framework and the rest of the analysis along several aspects. I give some more detail on this below, alongside other minor comments:

·       The authors use interchangeably food insecurity and FI throughout the paper. The same applies to US and United States and to community-based organizations and CBOs. I would suggest using only the acronym or the full name throughout the paper.

·       Lines 14-15: The phrase “at an academic medical center in collaboration with community partners” is not clear enough. Do the authors mean that the FFRx delivery program was developed by the academic medical centre in collaboration with community partners? Or that the interviewees attended the centre?

·       The sentences “The program promoted healthy dietary habits, improved access to high quality foods, improved wellbeing, enhanced financial wellbeing, and alleviated logistical barriers to accessing food and cooking” and “The FFRx program helped participants feel healthier and overcome financial and logistical barriers to healthy eating” are somewhat repetitive. I would suggest including only one which encompasses all the information.

·       Line 65: I would suggest adding here a brief note to explain what Figure 1 shows and how it works. Where is behaviour in the Figure? How are the different sources of behaviour, intervention functions and policy categories related?

·       Lines 68-70: The sentence “According to COM-B, to change behavior (i.e., healthy eating habits), one or more of the COM-B components would need to be changed or improved upon to help reduce barriers” could be improved. It is a bit contradictory to use “change behaviour” with “healthy eating habits” which is a positive attitude. I would suggest using something such as promoting more healthy eating habits in order to show in which direction change is desirable. In addition, it is not clear what the authors mean by the word "barriers". It has not been mentioned before.

·       Lines 68-76: I suggest giving an example involving the contents of Table 1 or Figure 1.

·       Line 77: Table 1 is entitled “Mapping of the COM-B System to TDF Domains and Strategies”.  I was wondering if this mapping has been specifically adapted to evaluating impacts of  dietary interventions. If so, I would suggest saying so.

·       Table 1: Table 1 is entitled “Mapping of the COM-B System to TDF Domains and Strategies”. I have a couple of issues here. First, no reference is made to behavior in the table. Second, what are TDF domains? There has been no reference to it until now. The title uses the words “TDF Domains” but the table “TDF Constructs”. This may lead to confusion. Third, the third column’s heading is named “Barriers/ Behavioural Modification Strategy”, but just as the barriers are clearly outlined, the Behavioural Modification Strategies are not. I would suggest making clear which are the strategies.

·       Lines 81-82: The text reads “This qualitative study was nested inside a larger mixed methods study exploring the impacts of a US produce prescription program on the health of participants”. Do the authors refer to this very same study or to a broader study? I would suggest being more specific on this.

·       Line 84: The text reads “to evaluate the impact of a community-informed FFRx program”. I would suggest saying “the community-informed FFRx program”. By now we all know the program.

·       Table 1 and Table 2: I would suggest using the same headings in both tables (the authors use “COM-B Domains” in Table 1 and “Capabilities-Opportunities-Motivation (COM-B) Framework Domains” in Table 2. One of the COM-B domains is missing in Table 2 and a not COM-B domain has been included. Why? Some TDF Constructs are not the same in Tables 1 and 2. If there is no specific reason for doing so, I would suggest using the same exact words to provide consistency.

·       Line 85: I would suggest briefly explaining exactly what Table 2 shows and relating the content to the COM-B methodology.

·       Line 89: I would suggest briefly explaining exactly what Table 3 shows.

·       Lines 92-100: This population refers to the one interviewed for the interviews, for the surveys, or both? I would suggest specifying so.

·       Line 124: I would suggest completing a sentence on “Table 2”.

·       Line 136: I would suggest completing a sentence on “Table 3”.

·       Line 142: This information “Surveys were conducted by telephone with English speaking program participants” has already been given in a previous sentence (in lines 139-140).

·       Lines 157-158: The text reads “survey responses related to the impact of the FFRx program”. I would suggest keeping to “quality improvement purposes” when speaking about surveys and “the impact of the FFRx program” when speaking about the interviews to avoid confusion.

·       Lines 162-164: I would suggest separating this sentence into a separate paragraph.

·       Lines 166-173: I would suggest explaining the characteristics of the survey participants too.

·       Line 173: I would suggest completing a sentence on “Table 4”.

·       Line 174: The authors only include here the number of interview participants. The number of survey participants is missing.

·       Lines 182-183: The text reads “(4) participants provided suggestions for program improvement”. Why is point number 4 not included in Table 5? In addition, I would like to know whether this point refers to the interview analysis of the survey analysis.

·       Line 184: I would suggest completing a sentence on “Table 5”.

·       Table 5: I would suggest writing “participants felt that the …” in “participants felt that the FFRx alleviated logistical barriers to food access” just as in the other themes or deleting the “participants felt that the …” from all points. In addition, I would use the word “wellbeing” instead of “health” in point 2 in the table (just as in the body of the text).

·       Lines 318-320: The sentence “This is the first study to use the COM-B model to both inform FFRx program development and qualitative interviews that complemented quantitative data in a mixed methods study” is not sufficiently clear. I would suggest rewriting it.

·       Line 338: What does “SDOH” stand for?

Round 2

Reviewer 1 Report

The authors noted that the research was based only on qualitative data [No quantitative data were collected as part of this study], clarifying how the results are fully expressed in a descriptive way.

Consistent with this assumption, the authors did not consider the requests for integration with quantitative data relevant, justifying it with “The quantitative data from this mixed methods study were previously published [18,19].

All other requests for corrections or additions to the text have been satisfied.

The revised text clarifies the initial assumption and has substantially improved the description and coherence of the research.

The deliberate choice to use only qualitative data inevitably undermines the scientific value of the paper, limiting the objectivity and consistency of the results. As a consequence of this methodological assumption, the paper, as the Authors point out, does not test hypotheses, rather, it generates hypothesis for future studies.

As for me, the paper can be published in the revised version